# Development of JavaScript-based deep learning platform and application to distributed training

**Masatoshi Hidaka, Ken Miura & Tatsuya Harada**
Department of Information Science and Technology
The University of Tokyo
7-3-1, Hongo, Bunkyo-ku, Tokyo, Japan
`{hidaka,miura,harada}@mi.t.u-tokyo.ac.jp`

## Abstract

Deep learning is increasingly attracting attention for processing big data. Existing frameworks for deep learning must be set up to specialized computer systems. Gaining sufficient computing resources therefore entails high costs of deployment and maintenance. In this work, we implement a matrix library and deep learning framework that uses JavaScript. It can run on web browsers operating on ordinary personal computers and smartphones. Using JavaScript, deep learning can be accomplished in widely diverse environments without the necessity for software installation. Using GPGPU from WebCL framework, our framework can train large scale convolutional neural networks such as VGGNet and ResNet. In the experiments, we demonstrate their practicality by training VGGNet in a distributed manner using web browsers as the client.

## 1 Introduction

Recently, machine learning, which uses big data derived from user activity on websites, images and videos is increasingly getting attention. Deep learning is at the center of that attention. Conventional machine learning techniques have required hand-crafted features specialized to a particular domain such as image or voice. In contrast, deep learning has a hugely important benefit that can illustrate data flow from raw data to an objective value in a single neural network and can train thoroughly using those data. In the computer vision domain, a team of Hinton (Krizhevsky et al., 2012) achieved outstanding classification accuracy using deep learning in an object classification competition ILSVRC2012 (Russakovsky et al., 2015). In the subsequent years' competitions, deep-learning-based methods evolved continually and exhibited superior performance (Simonyan & Zisserman, 2014a; Szegedy et al., 2014; He et al., 2016). Convolutional neural networks (CNNs) trained for ILSVRC object classification are helpful for improving classification accuracy for scene recognition and video recognition by functioning as a feature extractor or being fine-tuned (Zhou et al., 2014; Simonyan & Zisserman, 2014b). Moreover, application is beginning to emerge in other areas such as medical imaging (Tajbakhsh et al., 2016). Software platforms for deep learning are expected to play an important role in accelerating a wide range of research efforts and applications.

Although deep learning achieved significant recognition accuracy that cannot be achieved using conventional methods, the number of parameters that can be trained is greater, resulting in requests for huge amounts of training data. This shortcoming not only increases data collection costs but also increases computational costs of training larger parameters with larger data. Moreover, trial-and-error must be undertaken to ascertain a good neural network structure; thereby higher costs become necessary. What resolved this computational cost difficulty and enabled deep learning to work on a practical scale problem is general purpose computing on GPU (GPGPU) technology, which offers rapid matrix calculation. However, a deep learning framework must be set up on a dedicated computer. If a user wants to train a huge network, then a cluster computing system that uses MPI or Hadoop must be used for collaboration of multiple computers to obtain larger working memory and computational speed. To set up and maintain these systems generally presents

an expensive task. For that reason, such systems are available only to expert IT companies or laboratories.

This work specifically examines JavaScript, the programming language that runs on web browsers installed on ordinary personal computers and smartphones. With the recent advancement of web technology, JavaScript became the standard programming language to implement rich applications on web browsers. Word processors provided by Google and Microsoft are the popular examples. Those applications are traditionally implemented as native applications. This is not only a change of programming language; it brings an advantage of install-free convenience. Moreover, the communication features of web browsers are used not only during the loading of the application, but are also used by the application on demand, using so-called Ajax technology. For example, using this technology with a Google service spreadsheet, modifications made by one user are shown in real time on other users' displays. By making full use of this technology, collaboration of an application running on web browsers across the internet becomes possible. Moreover, web browsers such as Google Chrome run not only on Windows, but also on Mac OS X, Linux, Android, and iOS smartphones. They provide a compatible JavaScript executing environment. More recently, a small microcontroller board for prototyping Internet of Things (IoT) devices runs Linux. JavaScript can run on these devices. However, JavaScript is rarely used for scientific computation. This is mainly because JavaScript assumes single-threaded execution. It has no fast matrix computation library, which is crucially important for scientific computation. To resolve this difficulty, our previous work proposed the fast matrix computation library, which uses a parallel computing platform, WebCL, from JavaScript (Miura et al., 2015). In WebCL, GPGPU can be utilized from JavaScript code. Moreover, its application to deep learning is proposed (Miura & Harada, 2015). However, existing implementations cannot fully exploit the functionality of JavaScript and WebCL. For that reason, only a small six-layer CNN for classifying CIFAR-10 (Krizhevsky, 2009) dataset can be trained. In this work, our objective is to provide a deep learning platform that can train practical large-scale CNN as large as VGGNet. In the Experiment section, we present preliminary results on training VGGNet by distributed computation using web browsers as the computation client. In the following section, we restrict our description to CNN only, but our system is applicable to neural networks of other kinds by implementing the layers that they need.

Our contributions are the following:

- We implemented the fastest matrix library and deep learning library that can run on web browsers using GPGPU. The source code is provided as open-source software[1].

- Even where GPGPU cannot be used, native JavaScript implementation is provided, which allows high-level multi-dimensional matrix operation.

- We describe the possibility of training large scale CNN in a distributed manner without installing software in computation nodes, except for a generic plugin.

## 2 RELATED WORK

In this section, we first describe the studies related to distributed computing using generic computers that are not designed for scientific computing. The SETI@home project searches for extraterrestrial life (Anderson et al., 2002). In that research effort, radio waves analyses were performed distributedly on computers of volunteers. Although dedicated software had to be installed, more than 3 million computers participated in the project and contributed vast amounts of computational resources. Merelo-Guervos et al. (2008); Klein & Spector (2007) distributedly computed genetic algorithm (GA) using web browsers as computing nodes. The main component of GA was calculation of the fitness of population, which could be computed completely in parallel, thereby achieving extremely effective distributed computing. In our work, the main task to be distributed is deep learning, for which a large amount of weight parameters must be communicated frequently. Therefore, the communication efficiency becomes important.

Secondly, we explain distributed computing of deep learning. Dean et al. (2012) proposed a mechanism called DistBelief, which divides a neural network into multiple blocks of neurons and trains each block in a different computer. Large amounts of data are transferred at the division borders.

---

[1]Download code from `https://github.com/mil-tokyo`

They require n-to-n communication, which is unsuitable for environment in which computing nodes are not in the same LAN. deeplearning4j [2] provides distributed computing of deep learning framework that runs on the distributed computing Hadoop. However, Hadoop must be installed in all computing nodes, thereby imposing high deployment and maintenance costs. Meeds et al. (2014) developed a distributed deep learning system using web browsers. However, it is implemented in native JavaScript. For that reason, training with a large-scale dataset is nearly impossible because of the computational speed. In this work, we inherit the good properties of a JavaScript (web browser) based computing environment, with the aim of making training of practical CNN possible.

## 3    MATRIX LIBRARY IMPLEMENTATION

In this section, we describe the fast and generic matrix library "Sushi2", which is based on previous library "Sushi." They are using WebCL technology, which is a parallel computing platform to be used from JavaScript. WebCL is a JavaScript wrapper for parallel computing platform OpenCL, standardized by Khronos Group, which provides a unified interface to multi-core CPU and GPGPU. In contrast to NVIDIA CUDA, GPUs from AMD and Intel can also be used as accelerators. Unfortunately, WebCL is not built-in feature of web browsers, but there is an add-on for Firefox and WebCL-integrated Chromium. Our library also works with node.js (server-side JavaScript execution environment), in which node-opencl[3] library can be used to accelerate computation. Although Sushi2 performs best in a WebCL environment, most functions have equivalent native JavaScript implementation. Sushi2 currently uses WebCL for the acceleration of numerical calculation, but it is possible to use other solutions including WebGL or asm.js by substituting implementation of matrix manipulation. In WebCL, "kernel" is the function to run on GPGPU. Kernel, which is written in C language, must be compiled before use. Sushi2 wraps them to allow users to write simple codes. Details of low-level WebCL operations are available in the literature (Miura et al., 2015).

Though Sushi achieved efficient calculation on GPGPU, currently it lacks the availability for large scale neural networks that require matrices of large dimensions. Sushi2 is developed to overcome such problems that Sushi has been facing and achieved the following benefits:

- Use simple and efficient data structures to achieve good performance.

- Allow users to understand how to use it easily.

- Support CPU (native JavaScript) and GPGPU matrix without burdening ordinary users with learning WebCL programming.

Most general purpose matrix libraries for JavaScript represent a multi-dimensional matrix with a nested JavaScript array. In contrast, Sushi2 represents a matrix with TypedArray, which is used for transferring numeric data between the CPU and GPGPU. TypedArray is a one-dimensional numeric array with fixed size and bit width at construction, as in arrays of C language. The array accommodates efficient storing and manipulation of large data. TypedArray which stores 32-bit floating point numbers is named Float32Array and the one that stores 8-bit unsigned integer is named Uint8Array. The numeric type of JavaScript is a 64-bit floating point number, but some WebCL environments do not support it. Therefore, the basic numeric type of matrix is a 32-bit floating point number. However, the precision of a 32-bit floating number is only 23-bit, so it cannot be used as an index of a large matrix (which have more than $2^{23}$ elements). This is a problem for functions such as $argmax$, so a 32-bit signed integer matrix is also implemented. Moreover, an 8-bit unsigned integer matrix for raw image data and a logical matrix for Boolean operations are implemented.

Functions for the operating matrix are designed to be similar to those of MATLAB, which allows new users to use Sushi2 quickly. Operations for matrices that have more than two dimensions are implemented. It is a simple matter to operate color images and sets of color images (four-dimensional matrix). Almost all patterns for indexing operation in MATLAB are implemented. For import or export of a matrix, efficient binary format of numpy[4] is implemented as well as the native JavaScript nested Array.

---

[2] http://deeplearning4j.org
[3] https://github.com/mikeseven/node-opencl
[4] http://docs.scipy.org/doc/numpy/neps/npy-format.html

Table 1: Speed of Matrix Calculation. Time [ms] to process each task is shown.

- Task1: Addition of 1000x1000 matrix and 1000x1000 matrix
- Task2: Take element-wise logarithm of 1000x1000 matrix
- Task3: Multiplication of 1000x100 and 100x10 matrices
- Task4: Multiplication of 1000x1000 and 1000x1000 matrices

| Environment | Library | Task1 | Task2 | Task3 | Task4 |
|---|---|---|---|---|---|
| Firefox | Sushi2 + WebCL (Ours) | 15.6 | **12.8** | 33.6 | **62.4** |
| | Sushi2 (Ours) | **1.8** | 39.0 | **2.4** | 1897.8 |
| | Sylvester | 49.0 | 64.6 | 3.8 | 9438.6 |
| | Math.js | 36.2 | 503.4 | 16.0 | 23321.0 |
| node.js | Sushi2 + WebCL (Ours) | 4.0 | **14.0** | 3.8 | **5.2** |
| | Sushi2 (Ours) | **1.8** | 26.4 | **2.0** | 1891.0 |
| | Sylvester | 38.0 | 52.4 | 3.2 | 7102.8 |
| | Math.js | 53.8 | 679.2 | 19.8 | 57588.6 |

Function $M.gpuArray transfers a matrix to GPGPU. In functions that support WebCL, operations of matrices in GPGPU are accelerated. In JavaScript, unused memory is released by garbage collection, but this is not applied for memory allocated on the GPGPU by WebCL. It has to be released by explicitly calling the destruct method. To make programming convenient, an "autodestruct" helper function is supplied. When the closure passed to autodestruct finishes, the matrices allocated in it are released automatically. Figure 1 presents a sample implementation of a fully-connected layer of CNN. Whether input matrices are on GPGPU or not, they can be processed in the same code.

```
1  var top = $M.autodestruct(function () {// closure function
2      var product = $M.mtimes($M.t(weight), data);// weight' * data (No operator overloads in
           JavaScript)
3      var bias_repeated = $M.repmat(bias, 1, $M.size(data, 2));//$M.size(data, 2) is the number of
           samples
4      var product_with_bias = $M.plus(product, bias_repeated);// product + bias_repeated
5      return product_with_bias;
6  });// allocated matrices other than product_with_bias (e.g. $M.t(weight), product, bias_repeated) are
           released here
```

Figure 1: Example of forward calculation of fully-connected layer using Sushi2

Most GPGPU kernels are implemented originally for Sushi2, but matrix multiplication kernel is ported from clBLAS's[5] "sgemm", because it requires advanced optimization.

Table 1 presents a speed comparison between our library and existing JavaScript based matrix libraries; Sylvester[6] and Math.js[7]. The hardware environment is on Table 2 (AMD). When GPGPU is used, the time includes data transfer between the CPU and GPGPU. Task 1 represents simple element-wise task. Task 2 represents relatively expensive element-wise task. Task 3 and 4 are matrix multiplication task; the complexity of operations is greater than the number of elements. Our matrix representation (TypedArray) seems to be better than native JavaScript Array used in other libraries, even without WebCL. We can see clear superiority of using GPGPU when the computational cost is high.

# 4 DEEP LEARNING LIBRARY IMPLEMENTATION

In this section, we describe deep learning library "Sukiyaki2", which is based on matrix library Sushi2.

---

[5] https://github.com/clMathLibraries/clBLAS
[6] http://sylvester.jcoglan.com/
[7] http://mathjs.org/

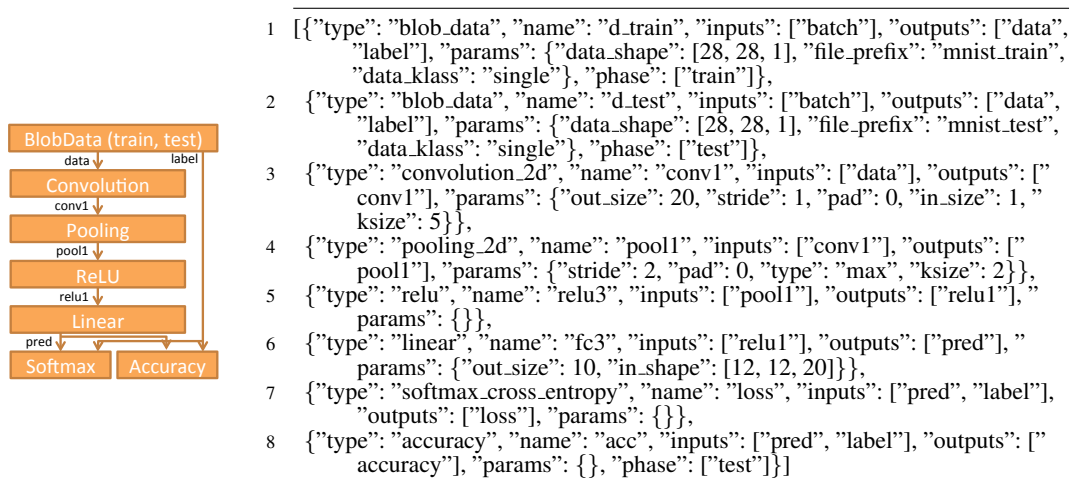

1 [{"type": "blob_data", "name": "d_train", "inputs": ["batch"], "outputs": ["data", "label"], "params": {"data_shape": [28, 28, 1], "file_prefix": "mnist_train", "data_klass": "single"}, "phase": ["train"]},
2 {"type": "blob_data", "name": "d_test", "inputs": ["batch"], "outputs": ["data", "label"], "params": {"data_shape": [28, 28, 1], "file_prefix": "mnist_test", "data_klass": "single"}, "phase": ["test"]},
3 {"type": "convolution_2d", "name": "conv1", "inputs": ["data"], "outputs": ["conv1"], "params": {"out_size": 20, "stride": 1, "pad": 0, "in_size": 1, "ksize": 5}},
4 {"type": "pooling_2d", "name": "pool1", "inputs": ["conv1"], "outputs": ["pool1"], "params": {"stride": 2, "pad": 0, "type": "max", "ksize": 2}},
5 {"type": "relu", "name": "relu3", "inputs": ["pool1"], "outputs": ["relu1"], "params": {}},
6 {"type": "linear", "name": "fc3", "inputs": ["relu1"], "outputs": ["pred"], "params": {"out_size": 10, "in_shape": [12, 12, 20]}},
7 {"type": "softmax_cross_entropy", "name": "loss", "inputs": ["pred", "label"], "outputs": ["loss"], "params": {}},
8 {"type": "accuracy", "name": "acc", "inputs": ["pred", "label"], "outputs": ["accuracy"], "params": {}, "phase": ["test"]}]

Figure 2: Sample of a neural network and corresponding definition file.

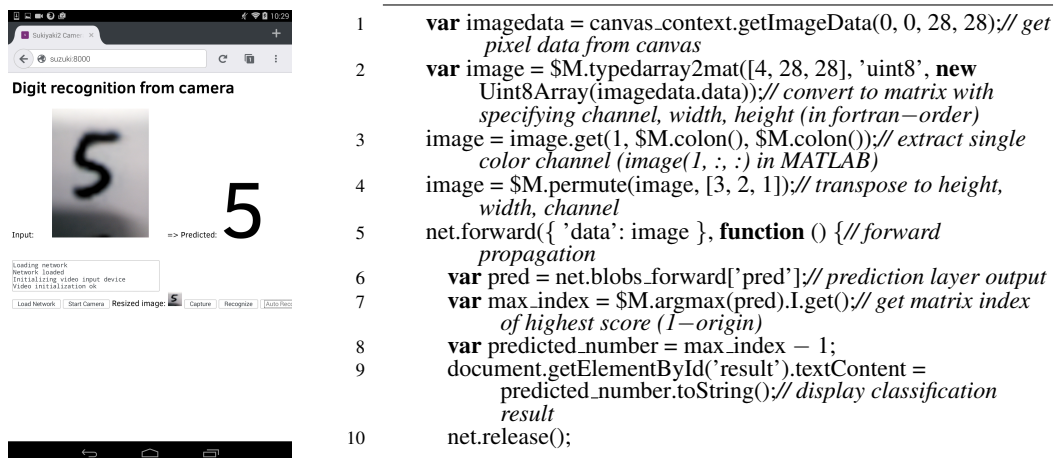

```
1   var imagedata = canvas_context.getImageData(0, 0, 28, 28);// get
        pixel data from canvas
2   var image = $M.typedarray2mat([4, 28, 28], 'uint8', new
        Uint8Array(imagedata.data));// convert to matrix with
        specifying channel, width, height (in fortran−order)
3   image = image.get(1, $M.colon(), $M.colon());// extract single
        color channel (image(1, :, :) in MATLAB)
4   image = $M.permute(image, [3, 2, 1]);// transpose to height,
        width, channel
5   net.forward({ 'data': image }, function () {// forward
        propagation
6     var pred = net.blobs_forward['pred'];// prediction layer output
7     var max_index = $M.argmax(pred).I.get();// get matrix index
        of highest score (1−origin)
8     var predicted_number = max_index − 1;
9     document.getElementById('result').textContent =
        predicted_number.toString();// display classification
        result
10    net.release();
```

Figure 3: Screenshot of digit recognition web application using trained CNN, and main code of recognition. Recognition is performed on Android tablet, not on server.

Sukiyaki2 implements modules that are necessary for deep learning: layers, network structure manager, and optimizers. Users can use a single layer separately, as well as training network by supplying configuration file to the executable. Figure 2 portrays a sample of a network definition file. For network analysis required for distributed computing in the future, we used the architecture with statically defined relations of layers. Improvements from our previous work include: enabling network graph branch (necessary for ResNet training), addition of some layers including dropout and batch normalization, efficient binary export of network parameters. Users can implement the original layers and optimizers to train new neural networks. It works automatically with CPU and GPGPU if it can be implemented by Sushi2's matrix operations. For cases in which a performance bottleneck exists, a dedicated GPGPU kernel can also be implemented. Using GPGPU for training is recommended, but almost all functions have native JavaScript fallback.

Figure 3 portrays a sample application for recognizing digits captured using a camera. The network is trained using MNIST dataset (LeCun et al., 1998b). Although image data are given as a flat byte array, extensive functions of Sushi2 allow short implementation of image recognition only in 10 lines. Recent web browsers for smartphones follow the JavaScript standard, and it is possible to develop such applications in this sample.

Table 2: Hardware used for the experiments. NVIDIA K80 is recognized as two independent GPGPU chips from software. Performance of the single chip is presented.

| GPU | GPU Theoretical FLOPS | CPU |
|---|---|---|
| AMD FirePro S9170 | 5.24T | Intel Core i7-5930K |
| NVIDIA K80 | 4.37T (using 1 chip) | Intel Xeon E5-2690 v3 |

Table 3: Speed of training LeNet. Processed images per second.

| JavaScript environment | ConvNetJS | Ours |
|---|---|---|
| Firefox | 64 | 107 |
| node.js | 88 | 4770 |

# 5 EXPERIMENTS

## 5.1 SINGLE-GPGPU TRAINING

In this section, we evaluate the CNN training performance of the proposed system. The specifications of hardware used for experiments are shown in Table 2.

First, we compared our library and existing deep learning library ConvNetJS by Andrej Karpathy[8], which is written in JavaScript. We evaluated them by training LeNet with MNIST dataset (LeCun et al., 1998b). The network structure is based on LeCun et al. (1998a), which contains two convolutional layers and two fully-connected layers. The batch size is 64. Firefox (version 32) and node.js (version 4.3.0) are used as the JavaScript execution environment. A tiny server application is implemented and used for supplying the dataset and saving the trained model to and from the web browser.

The measured calculation speed is presented in Table 3. In Firefox, the performance gain was relatively low because the control overhead of GPGPU is dominant in the small CNN. In node.js, this overhead is smaller, thus using GPGPU allowed faster computation by a large margin.

Next, we trained VGGNet (Simonyan & Zisserman, 2014a) and ResNet (He et al., 2016) as practical scale CNNs. VGGNet is proposed by Simonyan & Zisserman (2014a) at ILSVRC2014. 16-layer version, denoted as VGG16, includes 13 convolutional layers and 3 fully-connected layers. It is among the largest CNNs that are commonly used. ResNet is the winner of ILSVRC2015. 152-layer version, denoted as ResNet152, includes 151 convolutional layers and 1 fully-connected layer, but the bottleneck structure reduces the number of parameters.

We used Caffe (Jia et al., 2014), a popular deep learning library, for comparison. The mainstream version of Caffe employs NVIDIA CUDA as the interface to GPGPU. We designate this version as Caffe (CUDA). CUDA is not compatible with GPGPUs other than NVIDIA's. Caffe uses cuBLAS for matrix operations such as multiplication. There are forks of Caffe which use OpenCL as an cross-platform GPGPU interface. One such fork is OpenCL-Caffe by AMD[9], which uses clBLAS as the matrix operation. Another one is the opencl branch of Caffe by Fabian Tschopp[10]. It uses ViennaCL[11] for matrix operations. In Caffe (CUDA), the cuDNN accelerator library from NVIDIA can also be attached. We used same batch size in the same CNN / GPU setting for fair comparison.

The training speed is presented in Table 4. By virtue of GPGPU, VGG16 and ResNet152 can be trained, which was difficult using existing JavaScript based libraries. In ResNet152, more than 1,000 GPGPU kernels are executed and its execution overhead seems to be problematic on Firefox environment. Currently, our library is not faster than Caffe, but it achieved the same order of speed. Especially, Caffe (CUDA) provides the best performance. This difference mainly comes from the speed of convolution. Implementation of convolution in Caffe is similar to ours. To perform con-

---

[8]http://cs.stanford.edu/people/karpathy/convnetjs/index.html
[9]https://github.com/amd/OpenCL-caffe
[10]https://github.com/BVLC/caffe/tree/opencl
[11]http://viennacl.sourceforge.net/

Table 4: Training speed of VGG16 and ResNet152 [images/sec]. Batch size is shown in (). AMD represents AMD FirePro S9170, NVIDIA stands for NVIDIA K80.

| GPU | Software | VGG16 | ResNet152 |
|---|---|---|---|
| AMD | Ours (on Firefox) | 4.0 (32) | 1.4 (32) |
| | Ours (on node.js) | 5.7 (32) | 6.5 (32) |
| | Caffe (AMD) | 7.7 (32) | N/A |
| | Caffe (Tshopp) | 5.3 (32) | 1.6 (32) |
| NVIDIA | Ours (on Firefox) | 2.7 (16) | 0.2 (8) |
| | Ours (on node.js) | 4.9 (16) | 2.7 (8) |
| | Caffe (Tshopp) | 3.2 (16) | 1.5 (8) |
| | Caffe (CUDA) w/o cuDNN | 11.9 (16) | 8.5 (8) |
| | Caffe (CUDA) with cuDNN | 14.4 (16) | 9.4 (8) |

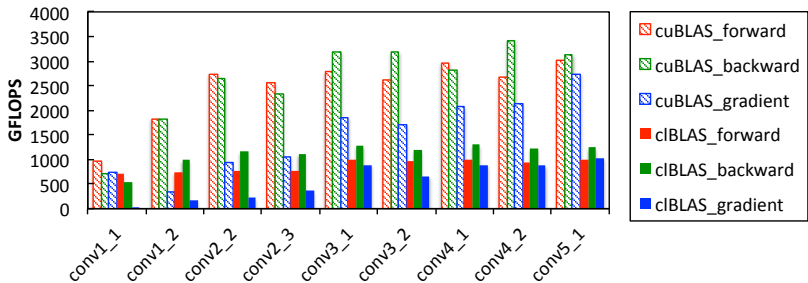

Figure 4: Calculation speed for each layer's computation in VGG16. Measured on NVIDIA K80 GPU. For example, forward computation of conv1_1 is performed by matrix multiplication of (802816, 27) and (27, 64). Forward, backward, gradient computation of cuBLAS and clBLAS are shown in different bars.

volution, elements of the input matrix are re-ordered (i.e. lowering). Then the output is gained by matrix multiplication with the weight. Table 4 presents the calculation speed in matrix multiplication used in computation of VGG16, performed by cuBLAS and clBLAS.

As the table shows, clBLAS gives inferior speed, especially on gradient computation of layers that are close to the input layer. In such layers, the matrix shape is far from square. For that reason, performance tuning for such input shape or implementation without matrix multiplication is needed. In the CUDA environment, Lavin (2015) showed that 96% of theoretical GPGPU performance is achieved in convolution by circumspect implementation.

## 5.2 DISTRIBUTED TRAINING

In this subsection, we describe a preliminary evaluation of distributed training of CNN.

The method of distributed training is simple data-parallelism. The system is depicted in Fig. 5. First the server distributes network weight $W_t$ and images in a batch. A batch for the iteration ($I_t$) is divided into $N$ splits, $I_{t1}, I_{t2}, ..., I_{tN}$, where $N$ is the number of computing clients. After the client $K$ calculates gradient of weight $\Delta W_{tK}$ using assigned batch split, they send the gradient to the server. The server takes the average of the gradients from all clients and then updates the weight using it ($W_{t+1} = W_t - \eta \frac{1}{N}\Sigma\Delta W_{tK}$). The optimization method is momentum SGD. The result is equivalent regardless of the number of clients.

First, we trained LeNet distributedly in Nexus 7 tablets (Android OS). Chrome browser is used as the client. The batch size is 120 and divided by the clients equally. Figure 6 (left) shows the speedup according to the increase in the number of clients. Naturally, the absolute speed is slow, but we can demonstrate that the computational power of mobile devices can be accumulated and nearly linear speedup is achieved.

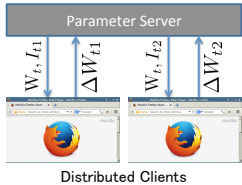

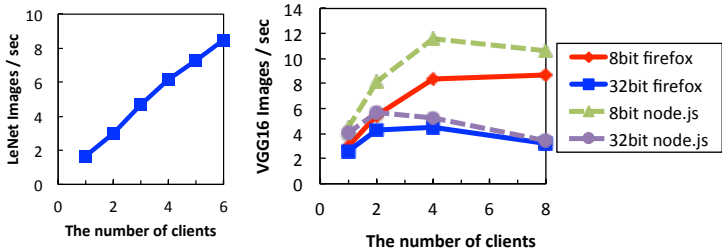

Figure 5: Data-parallelism system of distributed training

Figure 6: Computation speed with respect to the number of distributed clients. Left: speed of training LeNet in Nexus 7 Android tablets (Chrome browser). Right: speed of training VGG16 in clients with NVIDIA K80 (Firefox browser / node.js). Measurement includes time of communication and optimization in the server.

Next, we train large scale CNN; VGG16. Its weight and gradient have 130 million elements. It therefore requires 500 MB if represented as 32-bit floating point numbers, which poses a large communication bottleneck. To suppress this issue, we implemented 8-bit representation of each element proposed by Dettmers (2016). We used p2.xlarge instance of Amazon Web Services for GPGPU environment. It contains NVIDIA K80 GPU. The batch size is 256 according to (Simonyan & Zisserman, 2014a). Single forward-backward procedure cannot process 256 images at the same time due to the memory limit, so we average the gradients from multiple forward-backward procedure.

We show the speed of calculation with respect to the number of computing clients in Fig. 6 (right). Although our main focus is using web browser as clients, the result on using node.js as clients is also shown for reference. Under current settings, use of four clients achieved 2.8 times faster computation than with one client setting. The speed is much faster than existing OpenCL-based Caffe. Due to the communication overhead, the speed saturates at 8 clients even when 8-bit representation is employed.

Although we used K80, a high-end GPU, for this experiment, our motivation is to use ordinary personal computers for distributed computing. We can assume that latest ordinary personal computers (not dedicated for 3D game) have 1/10 performance compared to K80. In K80, we could train VGG16 with 29 seconds per iteration using 8 computers. In 1/10 performance GPU, we can estimate that maximum speed is 100 seconds per iteration using 16 computers, considering both calculation and network time. We compressed the weight to 1/4 size by the method of Dettmers, if we can compress it to 1/10 further, the maximum speed will be 31 seconds per iteration using 64 computers. Thus, further improvements demand reduction of communications and a better strategy of parallelism. We leave those improvements as a subject for future work.

## 6 CONCLUSION

We implemented a JavaScript based matrix library and deep learning library, to perform deep learning and to develop applications that use a trained model without a dedicated computer system. Using GPGPU via WebCL, our library provides much better performance than existing JavaScript based libraries. It became possible to train VGG16 and ResNet152. However, the performance is not reaching Caffe running on NVIDIA CUDA environment. A salient difficulty is that matrix multiplication necessary for convolution is slower. Additionally, we used WebCL as GPGPU interface, but currently it is not included in web browsers. Further improvements in web technology must be undertaken to make full computing power available to scripts in web pages. In experiments of distributed training of VGG16 using web browsers as computing client, 2.8x speed improvement was gained from four clients. The speed is much faster than existing OpenCL-based Caffe using single computer. The parallelization method used in the experiment is naïve, and further exploration of this area will be undertaken as a subject of future work.

ACKNOWLEDGMENTS

This work was supported by CREST, JST.

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
