# Peer review of "Development of JavaScript-based deep learning platform and application to distributed training"

_ICLR 2017 — rejected_

[Official Review · AnonReviewer3 · rating 4 · confidence 2 · 16 Dec 2016]
**Javascript wrapper for DNN code allows training within a web browser, even using GPU.**

While it is interesting that this can be done, and it will be useful for some, it does seem like the audience is not really the mainstream ICLR audience, who will not be afraid to use a conventional ML toolkit. 
There is no new algorithm here, nor is there any UI/meta-design improvement to make it easier for non-experts to design and train neural network systems. 

I think there will be relatively little interest at ICLR in such a paper that doesn't really advance the state of the art. 
I have no significant objection to the presentation or methodology of the paper.

[Official Review · AnonReviewer1 · rating 7 · confidence 3 · 16 Dec 2016]
**No Title**

This paper presents a JavaScript framework including WebCL components for training and deploying deep neural networks. The authors show that it is possible to reach competitive speeds with this technology, even higher speed than a compiled application with ViennaCL on AMD GPUs. While remaining a little more than factor three slower than compiled high performance software on NVIDIA GPUs, it offers compelling possibilities for easily deployable training and application settings for deep learning.

My main points of criticism are:
1. In Tab. 4 different batch sizes are used. Even if this is due to technical limits for the Javascript library, it would only be fair to use the smaller batch sizes for the other frameworks as well (on the GPUs probably in favor of the presented framework).

2. In Fig. 6, why not include more information in the graphs? Especially, as stated in the question, why not include the node.js values? While I do see the possible application with one server and many "low performance" clients, the setting of having a few dedicated high performance servers is quite likely. Even if not, these are good values to compare with. For the sake of consistency, please include in both subfigures Firefox, Chrome, node.js.

Apart from these points, well-written, understandable and conclusive.

[Official Review · AnonReviewer2 · rating 6 · confidence 4 · 24 Dec 2016]
**No Title**

Validity:
The presented work seems technically valid. Code for matrix library sushi2 and DL library sukiyaki2 are on github, including live demos that run in your browser.

[Public Comment · François Garillot · 14 Feb 2017]
**A small detail in the related work.**

In the § Related work:
"deeplearning4j 2 provides distributed computing of deep learning framework
that runs on the distributed computing Hadoop. However, Hadoop must be installed in all
computing nodes, thereby imposing high deployment and maintenance costs."

This is inexact, Deeplearning4j's most basic mode of operation is on a single machine, with Java installed. A GPU is used if available but is not a requirement (Deeplearning4j documentation:

[Final Decision · Program Chairs · 06 Feb 2017]
**ICLR committee final decision**

A summary of the reviews and discussion is as follows:
 
 Strengths
 Code for matrix library sushi2 and DL library sukiyaki2 are on Github, including live demos -- work is reproduceable (R2)
 Work/vision is exciting (R2)
 
 Weaknesses
 Projects preliminary (documentation, engineering of convolutions, speed, etc.) (R2)
 Perhaps not the right fit for ICLR? (R3) AC comment: ICLR specifically lists *implementation issues, parallelization, software platforms, hardware* as one of the topics of interest
 Doesn’t advance the state-of-the-art in performance (e.g. no new algorithm or UI/UX improvement) (R3)
 
 The authors responded to the pre-review questions and also the official reviews; they updated their demo and paper accordingly.
 
 Looking at the overall sentiment of the reviews, the extensive feedback from the authors, and the openness of the project I feel that it is a valuable contribution to the community. 
 
 However, given that the paper doesn't clearly advance the state of the art, the PCs believe it would be more appropriate to present it as part of the Workshop Track.